# Efficacy of *Artemisia annua* against Coccidiosis in Broiler Chickens: A Field Trial

**DOI:** 10.3390/microorganisms10112277

**Published:** 2022-11-16

**Authors:** Mircea Coroian, Loredana Maria Pop, Virgilia Popa, Zsuzsa Friss, Ovidiu Oprea, Zsuzsa Kalmár, Adela Pintea, Silvia-Diana Borșan, Viorica Mircean, Iustina Lobonțiu, Dumitru Militaru, Rodica Vârban, Adriana Györke

**Affiliations:** 1Department of Parasitology and Parasitic Diseases, University of Agricultural Sciences and Veterinary Medicine Cluj-Napoca, 400372 Cluj-Napoca, Romania; 2Pasteur Institute, Giulesti, 060269 Bucharest, Romania; 3The Research and Development Station for Cattle Breeding Târgu Mures, 547530 Sîngeorgiu de Mures, Romania; 4Oprea Avi Com, 547180 Crăieşti, Romania; 5Department of Microbiology, Immunology and Epidemiology, University of Agricultural Sciences and Veterinary Medicine of Cluj-Napoca, 400372 Cluj-Napoca, Romania; 6ELKH-ÁTE Climate Change: New Blood-Sucking Parasites and Vector-Borne Pathogens Research Group, 1078 Budapest, Hungary; 7Department of Chemistry, Biochemistry and Molecular Biology, University of Agricultural Sciences and Veterinary Medicine Cluj-Napoca, 400372 Cluj-Napoca, Romania; 8Department of Parasitology, Faculty of Biotechnology, University of Agronomic Sciences and Veterinary Medicine of Bucharest, 011464 Bucharest, Romania; 9Academy of Agricultural and Forestry Sciences Gheorghe Ionescu-Sisești, 011464 Bucharest, Romania; 10Department of Plant Culture, Faculty of Agriculture, University of Agricultural Sciences and Veterinary Medicine Cluj-Napoca, 400372 Cluj-Napoca, Romania

**Keywords:** *Artemisia annua*, artemisinin, broilers, chickens, *Eimeria*, coccidiosis, field

## Abstract

(1) Background: Various studies on artemisinin and its derivatives have shown that *Artemisia annua* may be of therapeutic interest for different diseases, including chicken coccidiosis. This study aimed to evaluate the effects of *Artemisia annua* on farm-reared broiler chickens by analyzing both the anticoccidial efficacy and its effect on the intestinal microbiota of poultry. (2) Methods: The experiment was performed within three houses on a broiler chicken farm located in Romania. House 1 was the experimental group and received a diet with an addition of *A. annua*. Houses 2 and 4 were the control groups and received anticoccidials. The prophylactic efficacy of *A. annua* against coccidiosis was evaluated by recording the weight gain, feed conversion rate, number of oocysts per gram of feces, lesion score, and mortality rate. (3) Results: The chickens fed with *A. annua* showed a decreasing trend in the number of oocysts per gram of faeces, and their lesion score was 80% lower than in the control group. The weight gains of the chickens treated with *A. annua* was lower, whilst the feed conversion rate was better than in controls. (4) Conclusions: *Artemisia annua* showed promising results in the prophylaxis of coccidiosis. Overall, the broiler chickens that received *A. annua* presented promising zootechnical performances and medical data related to coccidiosis and gut microbiota.

## 1. Introduction

Currently, the worldwide rise in drug resistance threatens the disease control of bacteria, viruses, fungi, or parasite-related human and animal infections. The emergence of drug resistance is due to both the misuse and overuse of antimicrobial substances, especially during disease prophylaxis of clinically healthy individuals. Humans can develop resistance to certain drugs by consumption of animal by-products treated with antimicrobial substances [1]. Thus, the World Health Organization (WHO) advocates for increased investment in the research and development of new drugs, vaccines, or diagnostic assays [2].

Poultry meat is expected to show the highest demand in the meat industry by 2050 with an overall increase of over 100% compared to previous years. Moreover, poultry meat plays an important role in improving nutrition in developing countries by providing a high-quality protein source and essential nutrients at a low price [3].

Avian-associated diseases are a major risk factor for maintaining meat production at requested levels. Coccidiosis is known as one of the most expensive diseases of the poultry meat sector, not only due to the high cost associated with prophylaxis and treatment but especially because of the major economic loss it generates by altering the productive performance of birds [4,5,6]. *Eimeria* species can lead to various clinical manifestations, thus making the overall impact of the disease difficult to estimate [7]. Although the disease is controlled in broilers mainly through chemoprophylaxis, the emergence of drug-resistant *Eimeria* strains and the growing consumer demands for organic products, emphasizes the need to find new natural molecules as alternative treatment [8].

The effective management of coccidiosis should consider the functionality, integrity, and overall gut health of poultry, which in turn also depends on many factors such as nutrition, environmental factors, or the gastrointestinal microbiota. The complex gastrointestinal microbiota of chickens is essential for the digestion and absorption of nutrients, pathogen removal, or the development of the immune system. Nonetheless, the diversity of the microbiota is influenced by the age of the birds, the location in the digestive tract, or even the diet [9].

Sweet wormwood (*Artemisia annua* L.) is a plant with annual growth from the Asteraceae family which can be found in the spontaneous flora of many countries such as China, Argentina, France, Bulgaria, Hungary, Romania, Spain, Italy, and the USA. [10]. The compound extracted from the leaves of *A. annua* is called artemisinin, which in combination with other synthetic drugs has proven to be the most effective treatment of malaria, a parasitic disease of major global importance [11]. To date, it has been shown that synergism between plant components does not reduce the rate of the recrudescence of malaria. However, compared to pure artemisinin, *A. annua* leaves can release 40 times more artemisinin into the bloodstream [12,13,14,15]. Recently, various studies on artemisinin and its derivatives have shown that this plant may also be of therapeutic interest for other diseases, including chicken coccidiosis [16,17].

This study aimed to evaluate the prophylactic efficacy of *A. annua* (German variety) leaf powder in broiler coccidiosis and the plant’s overall effect on the intestinal microbiota of the broilers in field conditions.

## 2. Materials and Methods

### 2.1. Study Design

The experiment was performed in a broiler poultry farm located in Mureş County (46°34′45.1″ N 24°36′31.3″ E), which is a large integrated farm (breeders, hatchery, broilers, and slaughterhouse) with a capacity of 2 million broilers/year. The farm technology is provided by Big Dutchman since 2016. Three broiler houses (H1, H2, and H4) were included in the study and assigned to experimental groups as follows: H1 (19,090 broiler chickens) was the *A. annua* experimental group (AaEG); H2 (17,250 broiler chickens) (CocCG H2), and H4 (18,900 broiler chickens) (CocCG H4) represented the anticoccidial control groups. Broiler chickens included in the study were the ROSS 308 hybrid. Broiler chickens from H1 and H4 originated from the same batch of parents (young breeders), while those from H2 came from a batch of old breeding parents. The choice of house H2 as a control was based on its proximity with experimental house H1 inside the farm, in order to avoid interferences generated by technological flow.

The broiler chickens were vaccinated against Newcastle disease on the 1st, 9th, and 23rd days of life (Avinew^®^, Merial; Nobilis ND Clone 30^®^, MSD Animal Health; Hipraviar^®^S, Hipra), and against infectious bursal disease on the 14th day of life (TA-bic^®^ M.B., Phipro Animal Health Corporation).

Coccidiosis prophylaxis in the control groups followed a shuttle program: Maxiban^®^ (narasin and nicarbazin, Elanco, 0.5 kg per 1 tonne of feed) from day 1 to day 21, and Elancoban^®^ (monensin sodium, Elanco, 0.5 kg per 1 tonne of feed) from day 22 to day 32. Coccidiosis prophylaxis in the experimental group was performed with in-feed *A. annua* leaf powder (AaLP) from day 1 to day 32. Anticoccidials and AaLP were introduced in the feed by the feed manufacturer (Agrifirm, Környe, Hungary). The efficacy of AaLP in the prophylaxis of broiler coccidiosis in comparison with anticoccidials was evaluated by recording parasitological and zootechnical performance parameters. Additionally, *Eimeria* species and gut microbiota were analyzed both in *A. annua* experimental and anticoccidial control groups (H2).

### 2.2. Artemisia annua Leaf Powder

*Artemisia annua* plant originated from the cultivar Anamed A-3 (Winnenden, Germany). Seeds were germinated on 16 March 2015 on peat support in a greenhouse, transplanted to the field on 25 May 2015 after gradual acclimatization, and harvested in early October 2015. *Artemisia annua* plants were cultivated in Sângeorgiu de Mureș (46°34′45.1″ N 24°36′31.3″ E) Romania, on the 1st terrace of the Mureș River. The soil is alluvial with a good supply of P_2_O_5_ (17.65 mg/100 g soil) and a medium supply of K_2_O (14.15 mg/100 g soil), a nitrogen index (IN) of 4.2 and a pH of 7.6, with a humus content of 3.94. After harvesting, plants were subject to artificial drying with warm air at 40–50 °C. Then, leaves were grounded to obtain AaLP. The artemisinin concentration in the leaf powder was analyzed by high-performance liquid chromatography (HPLC) analysis in 2015 after preparation, and in 2016 before the experimental study on the farm.

AaLP was analyzed qualitatively, microbiologically, and toxicologically, to assess the quality and safety of the product. All the analyses were performed in authorized laboratories from Romania (Sanitary Veterinary and Food Safety Laboratory from Cluj, Mureş, and Bistrița-Năsăud, and Institute of Hygiene and Veterinary Public Health, Bucharest).

The following constituents were evaluated in the qualitative analysis: raw protein (by Kjeldahl method), raw fat (by Soxhlet method), raw cellulose (by Reg (EC)152/2009 pt. I), and raw ash (by Reg (EC)152/2009-point M).

The microbiological analysis assessed the total number of germs (TNG by ISO 4833-1/2014), the bacterial species of the genus Salmonella (by ISO 6579:2003/AC2009), the number of Enterobacteriaceae (by ISO 21528-2:2007), and the number of yeasts and molds (by horizontal method, ISO 21527-2:2009).

During the toxicological analysis, the following mycotoxins were evaluated: T2 and HT2 toxins, total aflatoxins, ochratoxin A, fumonisin, zearalenone, and deoxynivalenol by competitive ELISA (RIDASCREEN^®^ kits, R-Biopharm, Germany).

The heavy metals tested were as follows: arsenic (BS EN 14546/2005), mercury (UNE EN 13806/2003), cadmium, and lead (BS EN 14082: 2003) by atomic absorption spectrometry and similar spectrometry.

Experimental broiler feed was prepared by the feed manufacturer by adding 3.5 kg of AaLP per ton of feed (0.35%) before extrusion.

### 2.3. Anticoccidial Efficacy Evaluation

The anticoccidial efficacy of AaLP was evaluated by recording the mortality rate (MR), the number of *Eimeria* spp. oocyst per gram of feces (OPG), lesion score (LS), body weight (BW) and body weight gain (BWG), and feed conversion ratio (FCR). These data were compared with data obtained in control groups.

The broiler chickens were monitored daily. Dead chickens were necropsied and recorded in the farm files. Approximately 500 g of fresh fecal droppings were collected at random by hand along the feed and water lines from the experimental and control groups (CocCG H2) on days 7, 14, 18, 21, 25, 28, and 35. First, the fecal samples were analyzed with the flotation technique using saturated sodium chloride (specific gravity 1.18–1.2) to detect *Eimeria* spp. oocysts. Positive samples were further analyzed by the modified McMaster technique to determine the OPG. Duplicate counts of duplicate fecal slurries were performed for each fecal sample.

The lesion score in broiler chickens was evaluated after the first oocysts were detected in the feces at the age of 28 days, and again one week later at 35 days-old. Ten chickens per group (AaEG, CocCG H2) were selected randomly from the house, and euthanized by cervical dislocation. Then, the whole intestinal tract of each broiler chicken was removed, opened, and specific *Eimeria* spp. lesions identified in the duodenum, jejunum, caeca, and the remaining parts of the gut. The lesions were scored from 0 to 4 according to the severity and scoring system of Johnson and Reid, 1970 [18].

Broiler chickens were weighed on the first day of life, and then weekly until slaughtering, and the BWG was calculated individually. The amount of administered feed to the broiler chickens was also counted for each house and the experimental group, and FCR was calculated at the end of the growing period.

### 2.4. Identification of Eimeria Species

Eimeria species were identified by qPCR using specific primers targeting SCAR markers derived from RAPD fragments for *E. acervulina, E. tenella*, and *E. necatrix*, and microneme protein gene 1 for *E. maxima* [19]. Oocysts from positive fecal samples were isolated, purified, and concentrated with a saturated salt solution [20]. Then, oocysts were washed three times with distilled water by repeated centrifugation [21]. Genomic DNA was extracted from 0.25 mg fecal sample and 100 μL oocysts pellets using the Isolate Fecal DNA (Bioline) kit according to the manufacturer’s instructions. The qPCR reactions were performed in the CFX96 Touch™ Real-Time PCR detection system (Bio-Rad, London, UK) using IQ Multiplex Powermix (Bio-Rad, London, UK) in a final volume of 20 µL. The amplification conditions were similar to the conditions described by Blake et al. [19] as follows: 1 × initial denaturation at 95 °C for 20 s; 40 × 95 °C denaturation, 15 s, and primer hybridization combined with 60 °C extension, 30 s. Data were collected at the end of each cycle.

### 2.5. Gut Microbiota

The impact of AaLP on the gut microbiota of broiler chickens was also assessed. For this purpose, samples from 10 broiler chickens belonging to each group (AaEG, CocCG H2 were collected by cloacal swab on the 1st, 14th, and 28th day of life. The analysis of the gut microbiota was performed by qPCR, based on the genetic sequences 16s ARNr for Enterobacteriaceae, Enterococcus, Bacteroides, Firmicutes, Bacteroidetes, and Eubacteria. To establish amplification parameters and detection limits for the qPCR, amplifications were performed with known concentrations of the DNA of the reference strains of *Escherichia coli* GDP4293 (for Enterobacteriaceae), *Enterococcus faecium* BM 4147 (for Enterococcus), *Bacteroides thetaiotaomicron* ATCC 29741 (for Bacteroides and Bacteroidetes), *Eubacterium rectale* ATCC 33656 (for Eubacteriaceae and Firmicutes).

Each swab with fecal samples was suspended in 400 μL NFW (Nutmeg Free Water, Promega P 3049), and a quantity of 200 μL of the sample was used for the DNA extraction. QiAamp DNA Minikit (Qiagen 51306, POS IP-CD-BM 268, the protocol for bacteria) was used for extractions. The extracted genomic material was quantified (Quant-it dsDNA High-Sensitivity Assay Kit, 0.2–100 ng, Life Technology Q 32851, with the Qubit fluorimeter, Invitrogen, POS IP-CD-BM 295), with 1 ng of DNA/sample being used in each amplification reaction. The amplification was performed in a final volume of 25 μL, which contained 12.5 pmol of primers (commercially synthesized Generi Biotech, Czech Republic) and the commercial mixture Brilliant II Sybr Green QPCR Master Mix (Agilent 600828), or Brilliant Multiplex QPCR Master Mix (Agilent 600553) for Bacteroides. The reactions were performed using the mx3005P spectrophotometric amplifier (Strata-gene/Agilent). The amplification programs used were as follows: (i) Enterobacteriaceae 50 °C, 2′ + 95 °C, 10′ − 1×; 95 °C, 15″ + 63 °C, 1′ − 40×; 72 °C, 10′ + 25 °C, 1′ − 1× [22]; (ii) Enterococcus 50 °C, 2′ + 95 °C, 10′ − 1×; 95 °C, 15″ + 61 °C, 1′ − 40×; 72 °C, 10′ + 25 °C, 1′ − 1× [22]; (iii) Bacteroides, Bacteroidetes, Eubacteria, and Firmicutes 50 °C, 2′ + 95 °C, 10′ − 1×; 95 °C, 15″ + 60 °C, 1′ − 40×; 72 °C, 10′ + 25 °C, 1′ − 1× [23].

The mean average number of genetic copies was calculated using the MxPro 4.10 software (Stratagene 2007/Agilent 2015). For graphic expression and analysis of the differences recorded between the two experimental batches of chickens, the obtained data were log-transformed (log10, MS Excel).

Moreover, the presence of avian pathogenic *E. coli* (APEC) strains was investigated by multiplex PCR targeting *ompA*, *iss*, and *fimH* genes using the protocol described by Popa et al. [24]. For this purpose, after the suspension of the swabs with fecal samples in sterile ultrapure water, they were immersed in BHI (Brain Heart Infusion) medium. The DNA was extracted from the resulting cultures.

### 2.6. Statistical Analysis

The arithmetic means, median, and standard error were calculated for OPG, and LS respectively. The normal distribution of data was assessed by the Kolmogorov-Smirnov test. Then, the *t*-test (independent samples *t*-test) was used to compare OPG between groups, and the Friedman test was used for LS. Differences were considered statistically significant if the *p*-value was ≤0.05. The data were processed with MedCalc software.

## 3. Results

### 3.1. Artemisia annua Leaf Powder

The artemisinin concentration in AaLP was 0.848% after harvesting (in 2015), and 1.706% one year later (2016) when it was introduced in the broiler chicken feed. The quantity of AaLP introduced in the feed was 3.5 kg (59.71 g artemisinin)/ton of feed (0.35%).

The results of the qualitative, microbiological, and toxicological analyses were according to the requirements of the in-force regulations (Table 1). AaLP had a crude protein content of 13.56% and 14.72% crude fiber. *Salmonella* spp. was absent. The toxins were under the admissible limit. Regarding heavy metals, only mercury was detected, but under the admissible limit.

### 3.2. Anticoccidial Efficacy

The weekly and total mortality rate for each house is presented in Table 2. The first week (X^2^_(1, 37,990)_ = 11.99, *p* = 0.0005) and total (X^2^_(1, 37,990)_ = 4.0243, *p* = 0.45) mortality rate was significantly lower (with 0.33%) in the *A. annua* experimental group (AaEG) than in the control group with chickens from the same batch (H4). When the total mortality rate was compared with the control from a different batch, AaEG had a higher rate (X^2^_(1,36340)_ = 4.2812, *p* = 0.039). However, the weekly and the total mortality rate of the AaEG and CocCG groups were in the range of the technological parameters for broiler chickens (Table 2).

*Eimeria* spp. oocysts were not detected in fecal samples until the age of 25 days in any house (Table 3). After this age, oocysts were identified in both, experimental and anticoccidial groups (H2). At the age of 25 (*t*_(6)_ = 48.328, *p* < 0.0001) and 28 (*t*_(6)_ = 10.888, *p* < 0.0001) days, the OPG value was significantly higher in chickens from the AaEG group. Later, at the age of 35 days, chickens fed with the *A. annua* diet shed 95.7% fewer oocysts (*t*_(6)_ = 33.928, *p* < 0.0001) compared to chickens fed with in-feed anticoccidial (CocCG H2).

Lesions caused by *Eimeria* spp. were found only in the duodenum (*E. acervulina*) irrespective of the group. In line with the OPG value, the LS was higher (with 37.5%) in the AaEG group at the age of 28 days (*p* = 0.17781), and lower (with 80.0%) at the age of 35 days compared with the CocCG (*p* = 0.07048) (Table 3). The statistically significant difference was noticed only inside the AaEG group between days 28 and 35 (*p* = 0.01613).

Day-one broiler chickens from the AaEG group had an overall lower BW compared to the control groups. The difference was 0.25 g between AaEG and CocCG H4 (same batch) groups, and 5.8 g between AaEG and CocCG H2 (different batch) groups, respectively. However, in the first three weeks of life, the AaEG broiler chickens had higher BW and BWG compared with broiler chickens obtained from the same batch of breeding parents (CocCG H4), and lower compared with broiler chickens obtained from a different batch of breeding parents (CocCG H2). Later on, broiler chickens from the CocCG groups were those with the highest values of BW and BWG. Thus, the broiler chickens from the AaEG group had a slaughter weight of 123 g, respectively, 207 g less than the broiler chickens from CocCG H2 (different batch) and H4 (same batch) groups (Table 4).

The best FCR was registered in the AaEG group. These broiler chickens consumed 36 g, respectively 47 g less feed/kg weight than broiler chickens from CocCG H4 (same batch), and CocCG H2 (different batch) groups (Table 4).

### 3.3. Identification of Eimeria Species

Samples with a Ct value <35 were considered positive. The qPCR analysis confirmed the infection with *E. acervulina*, a species that was identified during necropsy in both the experimental and the control groups. Additionally, *E. tenella* was also identified at the limit of detection (Ct = 30) in the samples from the AaEG group on day 35.

### 3.4. Gut Microbiota

It was observed that the qPCR results are more homogeneous in the AaEG group compared to the CocCG H2 group. The chickens that received the *A. annua* diet had a more homogeneous bacterial Bacteroides population than the coccidiostat group. Additionally, the melting temperature (T_m_) value of the AaEG group was slightly different from that of the reaction control and those of the CocCG H2 group. It may suggest that *A. annua* would induce point mutations in the bacterial genome (Table 5).

Samples collected from one-day-old chickens were APEC-negative. However, most of the samples (9/10) collected from 14- and 28-day-old chicken AaEG group recorded a full APEC profile. In contrast, in the CocCG H2 group on days 14 and 28, there were four and eight samples with full APEC profiles, respectively. The above-mentioned data points to a colisepticemia episode that might have evolved inside the farm with a causative agent resistant to both types of therapies (Table 6).

## 4. Discussion

Medicinal plants have been used since ancient times, gaining increasing popularity nowadays in particular due to the decrease in the effectiveness of synthetic substances and the public concern regarding adverse effects and drug interactions of synthetic molecules [10,25]. Natural herbal extracts have been intensively studied in recent years as new alternatives to traditional anticoccidials in the prophylaxis of chicken coccidiosis [26]. The extensive use of anticoccidials in the avian industry can lead to chemical residues in the meat and eggs. The consumption of products with chemical residues can cause allergic reactions, alteration of intestinal microbiota, and, most importantly, the proliferation of drug-resistant bacteria among the human population, ultimately leading to therapeutic failure [27]. Thus, the consumer’s interest in organic products has increased rapidly in recent years, and considering the global popularity of chicken meat the poultry industry was also faced with the growing demands of the public for organic products [28]. As a result, producers are forced to address this increasing requirement of the population while maintaining the safety and quality standards of the food.

The requirements of the official regulations regarding the certification of organic products restrict the use of synthetic molecules in the organic chicken-rearing system, therefore natural herbal products could be a viable solution for disease control [29,30]. Therefore, the present study aimed to assess the anticoccidial efficacy of *A. annua* in farm-reared broiler chickens.

*Eimeria acervulina*, a species responsible for reducing overall productive performances, was identified by both necropsy examination and qPCR in all three houses that were assessed, starting with the 25-day-old chickens. *Eimeria tenella*, a highly pathogenic species, which can be found in the cecum of chickens was also detected in the *A. annua* experimental house. This species was identified at the age of 35 days, at the limit of detection by qPCR, only from concentrated oocysts from fecal samples, and not directly from fecal samples. *Eimeria acervulina, E. tenella*, *E. maxima*, and *E. praecox* are the species reported in poultry farms from Romania [21]. In the study of Györke et al. (2013), *E. acervulina* was the only species identified in most of the large broiler farms. Ten years ago, the farm included in this study was a small-scale farm. At that time on the farm were identified *E. acervulina, E. tenella*, and *E. maxima*.

Our study showed a decreasing trend of OPG in broiler chickens from *A. annua* experimental group, while in the anticoccidial control group an increasing trend was noticed. This finding was also supported by the lesion score. At the age of 25 days, the OPG in AaEG was 81 times higher compared with CocCG H2; this evolution of OPG in the group treated with *A. annua* can be attributed to a weaker protection, which would have led to stronger immunity.

Almeida (2012) reported that the introduction of dried leaves of *A. annua* into the feed of chickens infected with *Eimeria* spp. led to a significant reduction in oocyst shedding by chickens [30,31]. The reduction of the OPG was also observed by Fatemi (2017) in chickens treated with an alcoholic extract of *A. annua* [32]. It seems that the anticoccidial effect of the plant resides in the action of artemisinin on the oocysts. Del Cacho (2010) concluded that artemisinin has a direct effect on the wall formation of the oocyst by inhibiting the SERCA expression in macrogametes, while Fatemi (2015), demonstrated that the alcoholic and petroleum ether extracts of *A. annua* inhibit oocyst sporulation due to morphological changes in the oocyst’s wall [33,34]. Additionally, artemisinin and *A. annua* leaves facilitate apoptosis of host cells and suppress the inflammatory response [35]. Moreover, Jiao et al. (2018) reported that *A. annua* leaves determined improvements in clinical symptoms, by promoting apoptosis and suppressing inflammatory response compared with artemisinin. Additionally, *A. annua* contains large amounts of flavonoids, tannins, and saponins that act as antioxidants and reduce oxidative stress caused by reactive oxygen species, a process that can also be seen in coccidiosis [36,37].

The percentage of mortality recorded in the *A. annua* experimental group was similar to that in the anticoccidial control groups and followed normal technological parameters. If in the first three weeks of life the BW and BWG were higher in broiler chickens from *A. annua* experimental group, in the last two weeks of life, these parameters were lower than in the control groups. Thus, the final weight at the slaughter was lower in the *A. annua* experimental group compared to the control groups. This decrease of BW and BWG in the second part of the study corresponded with high OPG counts on days 23 and 28 respectively and with the presence of *E. acervulina* that impairs the BW and FCR. Additionally, the fact that the broiler chickens from *A. annua* experimental house came from young breeders, could negatively influence the initial weight and feed consumption [38]. Engberg (2012) also reported a reduction in weight gain in chickens that were supplemented with a dry and ground *A. annua* product, but only at doses of 10 and 20 g plant/kg feed, while chickens that received 5 g/kg did not show any decrease in weight gain [39]. In the present study, a lower dose (3.5 g/kg feed) was administered in the feed of the experimental group, therefore the decrease in weight gain is not necessarily linked to the administration of the plant. Moreover, additional studies demonstrate the beneficial effects of the plant on productive performance [40,41]. Although the weight of the broiler chickens in the *A. annua* experimental group was slightly lower at the slaughterhouse, the FCR was superior compared with broiler chickens from the control groups. This result is also supported by the presence of a slightly different intestinal population in chickens from the experimental house. The bacterial groups *Bacteroides*, *Bacteroides*, *Eubacterium*, and *Firmicutes* were superior in number in the chickens from the experimental group compared to the chickens in the control group, while *Enterobacteriaceae* and *Enterococcus* were slightly inferior. The intestinal microbiota plays a key role in increasing nutrient absorption and strengthening the immune system, thus affecting both the growth and the health of chickens. As demonstrated in the human and swine species, *Firmicutes* help increase nutrient absorption. It was shown that an increased rate of *Firmicutes/Bacteroides* is closely linked to obesity in mice and humans [42].

## 5. Conclusions

*Artemisia annua* showed promising results in the prophylaxis of broiler chicken coccidiosis, in the context of the European directive regarding the use of antibiotics as growth promoters, but also for animal welfare. The broiler chickens whose coccidiosis prophylaxis was performed with *A. annua* presented promising zootechnical performances and medical data related to coccidiosis in comparison with those whose coccidiosis prophylaxis was done with Maxiban^®^/Elancoban^®^ (shuttle program). Additionally, *A. annua* had a positive effect on the gut microbiota. Supplementary research at the farm level in successive flocks is needed to gain a comprehensive understanding of the effects of using *A. annua* as a prophylaxis method.

## Figures and Tables

**Table 1 microorganisms-10-02277-t001:** Results of qualitative, microbiological, and toxicological analysis of *A. annua* leaf powder used in the study.

Type of Analysis	Parameter	Results	EU Regulation
Qualitative	Crude protein	13.56%	(EC) No. 152/2009
	Crude fat	2.88%	(EC) No. 152/2009
	Crude fiber	14.72%	(EC) No. 152/2009
	Crude ash	5.90%	(EC) No. 152/2009
Microbiological	TNG	1.2 × 10^5^ cfu/g	ISO 4833-1/2014
	*Salmonella* spp.	Absent/25 g	ISO 6579:2003/AC2009
	*Enterobacteriaceae*	3.1 × 10^3^ cfu/g	ISO 21528-2:2007
	Yeasts and molds	*Penicillium* and *Aspergillus* (4.2 × 10^4^ col/g)	ISO 21527-2:2008
Toxicological	T2 and HT2	0.0661 ± 0.0067 mg/kg	2013/165/EU (maximum = 0.1 mg/kg)
	Total aflatoxins	0.0069 ±0.0005 mg/kg	(EC) No 1881/2006
	Ochratoxin A	0.0181 ± 0.0011 mg/kg	(EC) No 1881/2006 (maximum = 0.25 mg/kg)
	Fumonisin	0.084 ± 0.007 mg/kg	(EC) No 1881/2006
	Zearalenone	0.0237 ± 0.0032 mg/kg	(EC) No 1881/2006 (maximum = 2 mg/kg)
	Deoxynivalenol	0.392 ± 0.03 mg/kg	(EC) No 1881/2006 (maximum = 8 mg/kg)
Heavy metals	Cadmium	Non-quantifiable value (<OQL)	BS EN 14082:2003
	Arsenic	Non-quantifiable value (<ODL)	BS EN 14546:2005
	Lead	Non-quantifiable value (<OQL)	BS EN 14082:2003
	Mercury	0.016 mg/kg	UNE EN 13806:2003

TNG: total number of germs; OQL: the quantification limit set in the laboratory; ODL: the detection limit set in the laboratory.

**Table 2 microorganisms-10-02277-t002:** The mortality rate in broiler chickens fed with *A. annua* supplemented feed in comparison with broiler chickens fed with anticoccidial supplemented feed.

	AaEG	CocCG			
	H1 *	H2	*p*	H4 *	*p*
Week 1	0.76	0.62	0.11	1.10	0.0005
Week 2	0.43	0.27	0.012	0.42	0.91
Week 3	0.40	0.26	0.022	0.30	0.13
Week 4	0.43	0.44	0.88	0.33	0.12
Week 5	0.40	0.50	0.16	0.48	0.20
Total	2.51	2.18	0.039	2.84	0.045

Legend: AaEG—*Artemisia annua* experimental group; CocCG—anticoccidial control groups; * Broiler chickens from these houses were from the same batch; *p*—statistical significance when AaEG group was compared with CocCG groups.

**Table 3 microorganisms-10-02277-t003:** Number of oocysts/g of feces (OPG) (average ± std. dev.) and total mean lesion score (TLS) (average ± std. dev.) in broiler chickens fed with the *A. annua* supplemented feed in comparison with broiler chickens fed with anticoccidial supplemented feed.

Groups	OPG			TLS (Duodenum)	
	D25	D28	D35	D28	D35
AaEG	2,029,635 ± 40,363 ^a^	1,393,605 ± 70,345 ^a^	29,304 ± 7992 ^b^	1.6 ± 0.89 ^a^	0.2 ± 0.45 ^a^
CocCG H2	24,975 ± 9565 ^b^	604,395 ± 17,489 ^b^	687,312 ± 17,671 ^a^	1.0 ± 0.0 ^a^	1.0 ± 0.71 ^a^

Legend: AaEG—*Artemisia annua* experimental group; CocCG—anticoccidial control groups; D—day of performing the analysis corresponding with the age of the chickens in days. Values with no common superscript in a column within an experiment were significantly different (*p* < 0.05).

**Table 4 microorganisms-10-02277-t004:** Body weight (BW) (g), body weight gain (BWG) (g/day), and feed conversion ratio (FCR) (kg feed/kg weight) in broiler chickens fed with *A. annua* diet compared with chickens fed with in-feed anticoccidial.

	AaEG (H1 *)		CocCG			
			H2		H4 *	
	BW	BWG	BW	BWG	BW	BWG
Day 1	39.9		45.7		40.15	
Week 1	150	15.72	185	19.9	150	15.69
Week 2	425	39.28	440	36.42	417	38.14
Week 3	943	74.0	1030	84.28	890	67.57
Week 4	1500	79.57	1600	81.42	1480	84.28
Week 5	2030	75.71	2200	85.71	2117	91.0
Total	2188	57.31	2311	62.82	2395	62.18
FCR	1.603		1.65		1.639	

Legend: AaEG—*Artemisia annua* experimental group; CocCG—anticoccidial control groups; * Broiler chickens from these houses were from the same batch.

**Table 5 microorganisms-10-02277-t005:** Average number of genetic copies of intestinal bacterial population in the two experimental groups evaluated by qPCR.

Group	AaEG			CocCG H2		
	D1	D14	D28	D1	D14	D28
*Bacteroides* *	1,633,720	1,413,220	252,097,000	1,812,030	1,061,830	10,088,846,400
*Bacteroidetes* *	67,982	44,165	7,732,450	44,173.8	192,737.8	2,278,603
*Enterobacteriaceae* *	72,572.1	8,440,100	13,356,900	1,948,194	11,022,681	14,524,884
*Enterococcus* *	107.987	7598.1	5293.787	1349.835	4317.125	22,270.14
*Eubacteria* *	0.113769	0.074434	7344.025	0.289981	0.052788	1475.237
*Firmicutes* *	195,320,000	17,774,600,000	18,016,100,000	525,596,000	12,168,776,100	14,927,400,000

Legend: AaEG—*Artemisia annua* experimental group; CocCG—anticoccidial control groups; D—day of performing the analysis that correspond with the age of the chickens in days. * Average number of genetic copies at three different ages.

**Table 6 microorganisms-10-02277-t006:** The number of positive samples using APEC genetic screenings in chickens treated with *A. annua* compared to chickens treated with anticoccidial.

	D1			D14			D28		
	ompA	iss	fimH	ompA	iss	fimH	ompA	iss	fimH
AaEG	0	0	0	10	9	10	10	9	10
CocCG H2	0	0	0	8	5	8	10	9	10

Legend: AaEG—*Artemisia annua* experimental group; CocCG—anticoccidial control groups; D—day of performing the analysis that corresponds with the age of the chickens in days.

## Data Availability

All data generated or analysed during this study are included in this published article.

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
