# Peer review of "Efficacy of *Artemisia annua* against Coccidiosis in Broiler Chickens: A Field Trial"

_microorganisms, 2022, doi:10.3390/microorganisms10112277_

Round 1
Reviewer 1 Report
This study evaluated the effects of Artemisia annua powder on farm-reared broiler chickens by analyzing both the anticoccidial efficacy and its effect on intestinal microbiota. The results are interesting and structure of the author's manuscript is reasonable, but there are several concerns should be solved before it could be published.
Major issues:
1. If the birds were experimentally infected with coccidia? If not, how could the comparation stands when chickens receive different dose of oocysts?
2. The result of this study is valuable because of its field experiments with large number of animals included. However, only ten birds were selected for the lesion scoring in representing the ~18k population in each group. And more details about the total lesion scores should be given as well as the statistical analysis.
3. Also about the lesion score. Line 161: “Lesion score was evaluated in 28, and 35 days-old broiler chickens”, why the two timepoints were selected to observe the lesion score?
4. Line 274-275: “However, in the first three weeks of life, the AaEG broiler chickens had the highest BW and BWG.”. From table 4, I can not see this, especially compared to the H2 group.
5. Table 5, there are so much numbers in same, please check it carefully.
6. If the author would like to present their data into graphics if it’s possible?
7. Animals are involved in this study, so a statement of ethics is required.
Minor issues:
1. Line 173-174, 290-291 and so on, E. acervulina, E. tenella, and E. necatrix and E. maxima should be written in italics.
2. OPG data of CocCG (H4) in table 3 is missing.
Reviewer 2 Report
1) They may perform comparative experiment between a new trial with A. annua and classical and/or typical methods, e. g., sulfas, before they concluded "a promising alternative" as an original article.
2) If the present MS is contributed as a short communication or research note as preliminary results because their results is relatively well, there is no problems.
3) The conclusion words "especially for organic farms" is showing as one of evident goals on this study. Hence, they had better show this in the introduction part of the MS. And, if possible, they may add that their studies will be contributed for the Animal Welfare as well to their MS.
Round 2
Reviewer 1 Report
All my questions are addressed.